# A coordinated progression of progenitor cell states initiates urinary tract development

Oraly Sanchez-Ferras [1], Alain Pacis[1,2], Maria Sotiropoulou[3], Yuhong Zhang[1], Yu Chang Wang[3], Mathieu Bourgey[2,3], Guillaume Bourque [2,3], Jiannis Ragoussis [3,4] & Maxime Bouchard [1✉]

The kidney and upper urinary tract develop through reciprocal interactions between the ureteric bud and the surrounding mesenchyme. Ureteric bud branching forms the arborized collecting duct system of the kidney, while ureteric tips promote nephron formation from dedicated progenitor cells. While nephron progenitor cells are relatively well characterized, the origin of ureteric bud progenitors has received little attention so far. It is well established that the ureteric bud is induced from the nephric duct, an epithelial duct derived from the intermediate mesoderm of the embryo. However, the cell state transitions underlying the progression from intermediate mesoderm to nephric duct and ureteric bud remain unknown. Here we show that nephric duct morphogenesis results from the coordinated organization of four major progenitor cell populations. Using single cell RNA-seq and Cluster RNA-seq, we show that these progenitors emerge in time and space according to a stereotypical pattern. We identify the transcription factors Tfap2a/b and Gata3 as critical coordinators of this progenitor cell progression. This study provides a better understanding of the cellular origin of the renal collecting duct system and associated urinary tract developmental diseases, which may inform guided differentiation of functional kidney tissue.

[1] Goodman Cancer Research Centre and Department of Biochemistry, McGill University, Montreal, QC, Canada. [2] Canadian Centre for Computational Genomics, McGill University, Montréal, QC, Canada. [3] Department for Human Genetics, McGill University Genome Centre, McGill University, Montréal, QC, Canada. [4] Department of Bioengineering, McGill University, Montreal, QC, Canada. ✉email: maxime.bouchard@mcgill.ca

**K**idney development proceeds by reciprocal interaction between two main tissues. The epithelial ureteric bud (UB) branches repeatedly to eventually form the collecting duct and ureter of the adult kidney (the metanephros), carrying urine down to the bladder[1–3]. In turn, ureter tips induce the formation of nephrons from progenitor cells located in the surrounding mesenchyme. Mature nephrons filtrate circulating blood and produce urine by reabsorption of important nutrients. In recent years, considerable progress has been made to understand the identity of nephron progenitor cells[3–15]. In contrast, little is known about the developmental origin of the UB, the other major component of the definitive kidney.

The UB is known to emerge from the nephric duct (ND), the central component of the pro/mesonephros (Fig. 1a). This primitive kidney starts with the formation of an epithelial duct (the ND) from intermediate mesoderm cells of mid-gestation embryos (embryonic day (E)8.75). The ND elongates by collective cell migration over 24 h to reach and fuse with the cloaca (primordium of the bladder, urethra and hindgut)[16–23] (Fig. 1a). During duct elongation, mesonephric tubules (MT) are induced in the adjacent intermediate mesoderm by signals from the ND[24]. These tubules are primitive nephrons that later differentiate into efferent ducts of the male genital system[24,25], while the ND becomes part of the male genital tract[26,27]. Hence, the sequence by which the UB derives from the ND and the ND from the intermediate mesoderm is well established at the morphological level, but the identity and heterogeneity of the cells involved in this process are currently unknown.

Consistent with the central role played by the ND primordium in genitourinary tract development, defects in ND morphogenesis lead to congenital anomalies of the kidney and urinary tract (CAKUT), a complex disease group that accounts for 50% of chronic renal failure cases in children and about 20–30% of all congenital anomalies detected prenatally[28–31]. Mouse and human genetics approaches have uncovered a complex gene regulatory network involved in urinary tract morphogenesis, which includes essential ND and CAKUT-related genes such as *Pax2*, *Gata3*, *Ret*, *Lhx1* and *Emx2*[32–43].

Here, we present several lines of evidence that the ND/UB lineage precursor develops by sequential generation of four major ND progenitor cell populations. We further define *Gata3* and *Tfap2a/2b* as critical regulators of the progenitor cell progression initiating urinary tract development.

## Results
### Single-cell profiling of *Pax2* expressing progenitor cells. To address cell heterogeneity in early renal development, we performed single cell RNA-seq of *Pax2*-expressing cells isolated by Fluorescence-activated cell sorting (FACS) from the trunk of E9.5 *Pax2-GFP* BAC transgenic embryos (Fig. 1a,b and Supplementary Fig. 1)[44]. At this developmental stage, *Pax2-GFP* marks renal cells (ND and MT) as well as other cell types including intermediate mesoderm and tailbud[45–47].

The experiment yielded a total of 3396 cells, which clustered into 7 cell populations identified as ND, MT, intermediate mesoderm, tailbud, hindgut, pharyngeal ectoderm and neural (Fig. 1b). ND cells ($n = 1060$), were defined as *Pax2*+, *Gata3*+, *Wt1*−, whereas mesonephric tubule cells ($n = 723$) were defined as *Pax2*+, *Gata3*−, *Wt1*+. In total, renal cells accounted for 53% of all cells (31% ND identity) and clustered closely to intermediate mesoderm cells (*Col1a2*+), consistent with the developmental relationship between these cell types (Fig. 1b). Based on the top-ranking differentially expressed genes between those populations, we derived gene signatures for ND, MT, intermediate mesoderm and tailbud *Pax2*-positive cell lineages from E9.5 mouse embryo

(Fig. 1c). This analysis of *Pax2*-expressing cells in the trunk region highlights lineage relationships between mesoderm-derived cell populations and defines clear renal cell populations to be studied further.

### The renal primordium harbors distinct progenitor populations. Unbiased clustering analysis of ND cells (*Gata3*+, *Wt1*−) by tSNE (t-distributed stochastic neighbor embedding) dimensionality reduction identified four subclusters. We named these Nephric duct Progenitor (NdPr)1, NdPr2a/b, NdPr3 and NdPr4 (Fig. 2a and b, Supplementary Fig. 2a).

The NdPr1 cell population (defined as *Gata3*+, *Hoxb9*$^{high}$ *Notch2*$^{low}$ *Tfap2b*$^{low}$, *Aldh1a3*$^{low}$) expresses several transcription factor genes (e.g., *Pax8*, *HoxA/B/C genes*, *Cited2*, *Uncx*), as well as the signaling molecules *Sfrp2*, *Pcp4* and *Fgf8* (Fig. 2c NdPr1 panel, Supplementary Fig. 2a). Notably, most of the genes enriched in NdPr1 are not exclusive of the ND lineage but are also expressed in intermediate mesoderm and MT (Fig. 1c, Fig. 2e NdPr1 panel, Supplementary Fig. 4). To trace the location of NdPr1 cells in the ND, we performed marker gene expression on ND tissue sections. We divided the E9.5 ND along the rostro-caudal axis into four regions: Rostral, Intermediate I, Intermediate II and Tip (Fig. 2e). Each of these regions exhibit different morphologies and behavior during the ND elongation process[20] (Supplementary Fig. 3a and b). Immunostaining for Hoxb9 shows that NdPr1 cells mostly localize to the anterior ND (Rostral and Intermediate I). A time course analysis of embryos between E8.75 and E9.5 further detected Hoxb9 expression in the early pro/mesonephros (E8.75), which progressively decreases during ND development (Fig. 2e NdPr1 panel).

The NdPr2 cell population (*Gata3*+, *Hoxb9*$^{low}$, *Notch2*$^{low}$, *Tfap2b*$^{high}$, *Aldh1a3*$^{low}$) is characterized by the downregulation of NdPr1 markers and upregulation of the transcription factor *Tfap2b* (Fig. 2c and d NdPr2 panel, Supplementary Fig. 2a). This population is found in two clusters (NdPr2a and NdPr2b) characterized by different levels of *Sfrp2* (downregulated) and *Tfap2b* (upregulated) (Fig. 2c and d NdPr2 panel, Supplementary Fig. 2a). Other NdPr2 markers include the actin regulator *Gsn*[48] and the transcriptional co-activator *Cited1* (Fig. 2c NdPr2 panel, Supplementary Fig. 2a). Histological validation with an anti-Tfap2b antibody shows an expression restricted to the rostral and intermediate regions of the ND, while excluded from the ND tip region and mesonephric tubule cells at E9.5 (Fig. 2e NdPr2 panel). Tfap2a, another Tfap2 family member and ND marker (Fig. 1c), showed a similar pattern of expression as Tfap2b (Supplementary Fig. 3b). Time course analysis revealed that *Tfap2b* expression in the ND is turned on at E9.0 and maintained in the rostral region at E9.5 (Fig. 2e NdPr2 panel).

The NdPr3 cluster (*Gata3*+, *Hoxb9*$^{low}$, *Notch2*$^{high}$ *Tfap2b*$^{high}$, *Aldh1a3*$^{low}$) maintains high *Tfap2b* expression but additionally express the signaling molecules *Jag1*, *Notch2*, *Spry2* (Fig. 2c and d NdPr3 panel, Supplementary Fig. 2a). Histological validation of NdPr3 using an anti-Notch2 at E9.5 showed an enrichment in a subset of ND cell in the vicinity of MT (Fig. 2e NdPr3 panel). At E10, Jag1, Notch2 and Kdelr2 confirmed an enriched expression of NdPr3 markers in the mesonephric connecting segment (CnS) located between ND and mesonephric tubule cells (Supplementary Fig. 3c).

The NdPr4 population (*Gata3*$^{high}$, *Hoxb9*$^{low}$, *Notch2*$^{low}$, *Tfap2b*$^{low}$, *Aldh1a3*$^{high}$) shows a downregulation of NdPr1, NdPr2 and NdPr3 markers, notably *Hoxb9* and *Tfap2b*, and upregulation of *Aldh1a3* and *Wnt6* (Supplementary Fig. 2a). Among other specific markers of this population are the known regulators of ND elongation and UB development (i.e., *Ret*, *Npnt*, *Robo2*, *Gfra1*, *Plac8*)[32,33,35,49–51], (Fig. 2c and d NdPr4 panel) as

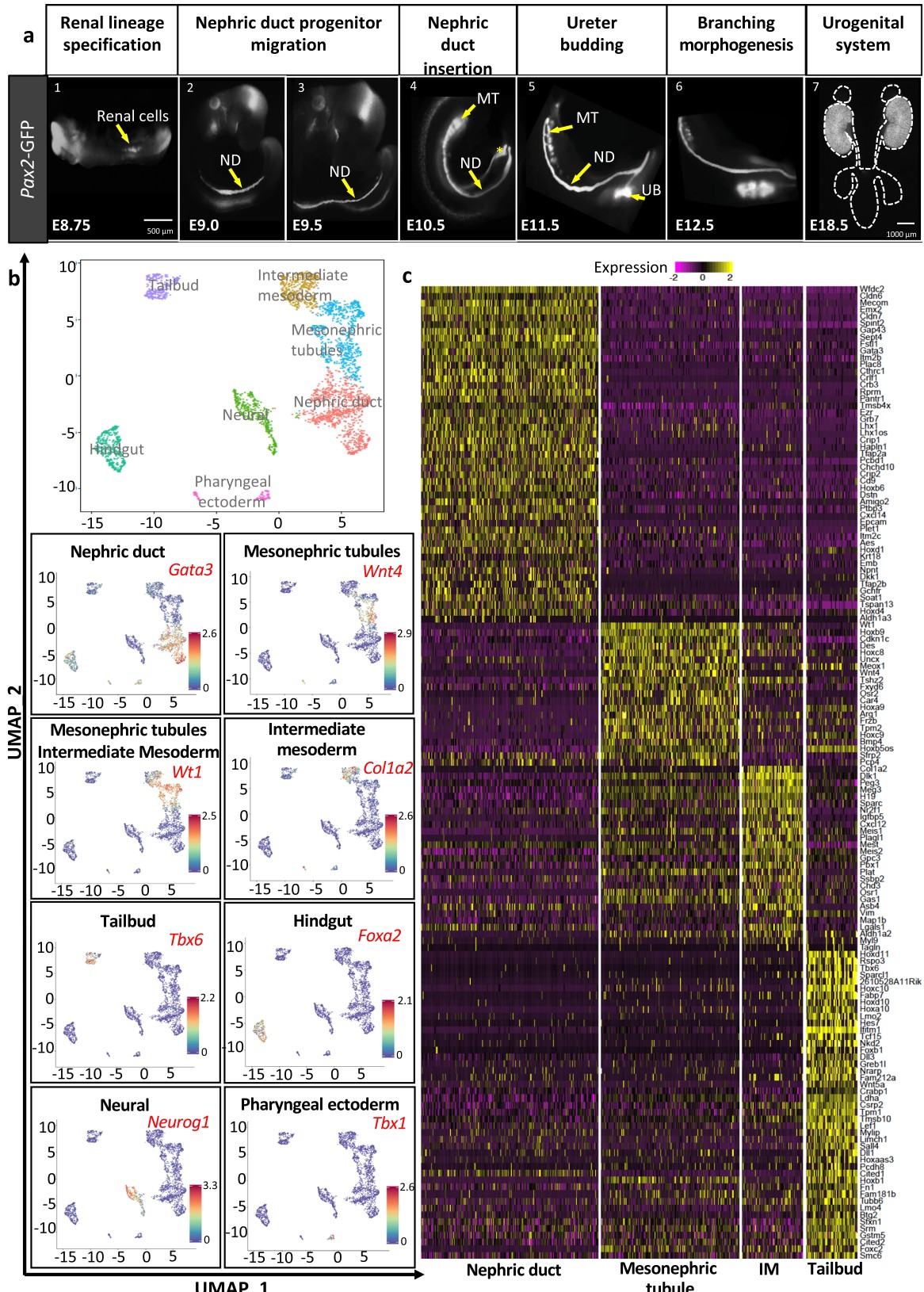

well as previously identified markers of UB lineage maturation such as *Tbx2*, *Epcam* and *Calb1*[5]. One of the most enriched Gene Ontology biological processes associated with NdPr4 is UB morphogenesis (Supplementary Fig. 2b). The spatial and temporal analysis for this population using immunofluorescence against Aldh1a3 revealed that NdPr4 is the last cell population

that emerges in the ND, detected at the protein level from E9.5 on (Fig. 2e NdPr4 panel). Importantly, NdPr4 is a caudal population restricted to the Intermediate II and Tip regions of the developing ND (Fig. 2e NdPr4 panel).

A similar exercise with the mesonephric tubule lineage (*Gata3⁻*, *Wt1⁺*) identified three main subclusters: MtPr1,

**Fig. 1 Single-cell RNA-seq of Pax2-GFP positive cells identifies related cell populations in the caudal trunk. a** Representative wholemount GFP fluorescence images of *Pax2-GFP* embryos highlight urogenital system development at different stages. The nephric duct undergoes specification and migration along the rostro-caudal axis of the embryo. Caudal nephric duct cells later form the collecting duct lineage of the metanephros. $n = 10$ embryos per developmental stage. Scale bar for pictures 1–6 is 500 µm, scale bar for picture 7 is 1000 µm. **b** Uniform Manifold Approximation and Projection (UMAP) analysis identified 7 *Pax2*-positive cell populations in the caudal trunk. Expression of selected cluster markers is shown in the lower panel. **c** Expression heatmap of relevant cluster markers across *Pax2-GFP* positive nephric duct, mesonephric tubules, intermediate mesoderm and tailbud cells. Expression values presented as Z-scores. Shown are cluster-defining genes, identified using a logFC > 0.25 and adjusted P_value < 0.05. ND: nephric duct, MT: mesonephric tubules, IM: intermediate mesoderm, UB: ureteric bud.

MtPr2 and MtPr3 (Supplementary Fig. 4). The MtPr1 cell population (*Ccnd1*High, *Wt1*low, *Notch2*low) also expressed *Lhx1, Fgf8 and Wnt4* and corresponds to epithelializing MT equivalent to comma-shaped bodies of the metanephros[24,52] (Supplementary Fig.4). MtPr2 (*Ccnd1*low, *Wt1*high, *Notch2*low) highly expressed the intermediate mesoderm marker *Osr1*, as well as *Vimentin* (*Vim*) and likely represent a transition stage between intermediate mesoderm and mesonephric tubule (Supplementary Fig.4). Finally, MtPr3 (*Ccnd1*low, *Wt1*+, *Notch2*high) showed similarity in marker gene expression with NdPr3 (Pearson correlation coefficient $r_{NdPr3-MtPr3} = 0,604$). This lineage may represent progenitors of the CnS (Supplementary Fig. 4).

Together these results identify distinct progenitor cell populations in the ND and mesonephric tubule lineages of the renal primordium.

**Cluster-RNA-seq supports a spatial segregation of NdPr.** The expression pattern of NdPr1-NdPr4 at E9.5 suggests that there is a spatial segregation of cell lineages along the rostro-caudal axis of the elongating ND (Fig. 2e). To assess the spatial heterogeneity of the different cell populations (Fig. 3a), we developed a strategy called Cluster RNA sequencing (cl-RNAseq). Though similar to single cell (sc)-RNAseq, this method uses small cell clusters (4–6 cells) instead of single cells as input for RNAseq libraries. *Pax2-GFP* positive cell clusters were generated by partial trypsinization and FACS sorting for large-size GFP positive units (Fig. 3b and Supplementary Fig. 1). This technique captures clusters of cells that are physically connected, thereby providing a measure of 3D cell neighboring. Importantly, cl-RNAseq is performed on the same equipment as sc-RNAseq, allowing for spatial transcriptomics information to be obtained in parallel to sc-RNAseq data without the need for additional specialized technology characteristic of other spatial transcriptomics approaches[53–58].

Cluster and single cell libraries were filtered based on the number of unique molecular identifier (UMI), resulting in libraries containing a mean of 6 cells per cluster (Supplementary Fig. 5a). The resulting libraries generated 6 clusters (Cl-a to Cl-f) representative of all four major ND cell populations (Fig. 3c, Supplementary Fig. 5b-c). Pearson correlation analysis found association between Cl-a, Cl-b and Cl-d with NdPr4, NdPr1 and NdPr3, respectively ($r > 0.6$ and random permutation $p < 0.01$) (Fig. 3c, Supplementary Data 1). Interestingly, Cl-c, Cl-e and Cl-f harbored mixed NdPr identities (NdPr1/2; NdPr2/3 and NdPr2/4, respectively), (Fig. 3c and Supplementary Data 1). This cell cluster analysis indicates that NdPr2 cells can neighbor all ND progenitor types, thus suggesting a central position in NdPr progression. Conversely, NdPr1 and NdPr3 are never found in physical proximity to NdPr4, consistent with NdPr1 being associated with NdPr2, which is itself associated with either NdPr3 or NdPr4 (Figs. 2e and 3c-e). To validate the Cluster-seq results, we performed spatial transcriptomics[57] using the Visium expression 10× protocol on E9.5 ND. This transcriptomic analysis of small spots extracted from the rostral region of the ND (about 10 cells per spot) confirmed the enrichment and proximity of

NdPr1, NdPr2 and NdPr3 (found in cluster subgroups b, c, d and e) but not NdPr4 (found in cluster subgroups a and f) (Fig. 3d and Supplementary Data 1, Supplementary Fig. 5d).

Taken together, these results introduce and validate Cluster-RNA-seq as a simple and robust approach to infer 3D neighbor analysis from RNA-seq data that can be done in parallel with scRNAseq. Together, Cluster-RNA-seq and Visium spatial transcriptomics suggest a defined organization of NdPr cell populations along the rostro-caudal axis of the ND (Fig. 3e).

**Temporal emergence of NdPr progenitors.** To specifically address the temporal emergence of NdPr cell populations, we performed single cell RNA-seq analyses on FACS sorted renal cells at different time points of ND/UB development (E8.75, E9.0, E9.5 and E11.5) (Fig. 4a, Supplementary Fig. 1b and 6). This experiment showed an enrichment of anterior intermediate mesoderm and NdPr1 markers such as *Osr1* and *Pcp4* in E8.75 and E9.0 ND cells (Supplementary Fig. 7a). NdPr1 accounted for over 80% of total ND cells at these early stages and progressively decreased at later stages (Supplementary Fig. 7a and 7b). Of note, expression of some NdPr1 markers such as *Sfrp2* was already present in the anterior intermediate mesoderm of E8.5 embryos[59], supporting the idea that NdPr1 cell represent a transition state between anterior intermediate mesoderm and NdPr2 ND cell fates ([59] and Supplementary Fig. 7c, d). NdPr2 cells first appeared at E9.0 and became the most abundant ND cell type at E9.5, accounting for near half of total ND cells (Supplementary Fig. 7a and 7b). A small number of NdPr3 cells were detected at E9.0, but this cell type was more abundant at E9.5 (18%) (Supplementary Fig. 7a and 7b). Finally, NdPr4 emerged only at E9.5 accounting for 11% of total ND cells (Supplementary Fig. 7a and 7b). Importantly, NdPr4 was the ND cell type that clustered closest to E11.5 UB cells in UMAP (Fig. 4a and Supplementary Fig. 7a). To confirm this fate proximity, we performed wholemount immunostaining with both Aldh1a3 (NdPr4) and Tfap2b (NdPr2) at E10.5 when the UB is just emerging. This experiment revealed the presence of NdPr4 but no NdPr2 in the region forming the UB (Supplementary Fig. 7e).

To construct the developmental trajectory of NdPrs, we analyzed ND cells collected at successive developmental stages by pseudotime using the Monocle 2 toolkit. This analysis placed NdPr1 at the start point, together with E8.75 and E9.0 ND cells (Fig. 4b and Supplementary Fig.8). NdPr1 cells are closest to NdPr2, which branches into two distinct cell types, NdPr3 and NdPr4. Here again, NdPr4 was found together with E11.5 UB cells, suggesting a close lineage relationship (Fig. 4b,c and Supplementary Fig. 8). This data supports the proposed lineage progression in which NdPr1-4 cell populations emerge in a temporal manner during ND development.

To validate the proposed lineage relationship, we assessed the developmental potential of NdPr2 cells in vivo by performing tissue transplantation assays (Fig. 4d). For this, we finely dissected NdPr2 enriched tissue from the intermediate ND of E9.5 *Pax2-GFP* embryos, devoid of the region containing MT, mesonephric

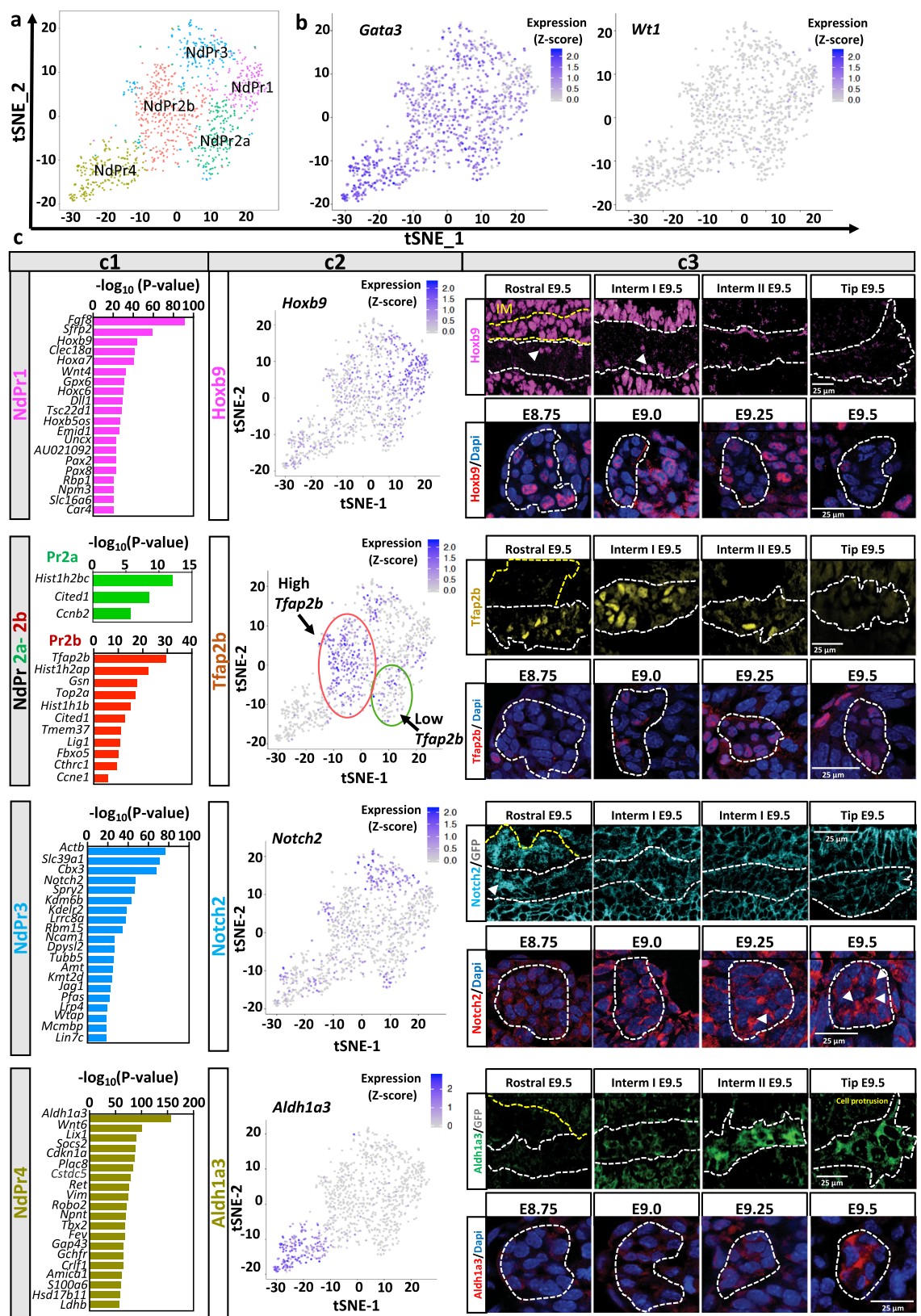

connecting duct cells and the Aldh1a3+ cell population (Fig. 4e and f, $t = 0$ h). NdPr2 expressing grafts were implanted near the rostral or caudal ND (Fig. 4d, e, f) of wild type embryos to assess their differentiation potential. After 24 h in culture rostral grafts acquired a NdPr3 fate (Fig. 4e). Conversely, NdPr2 explants grafted in the caudal trunk downregulated Tfap2b expression and turned on Aldh1a3 expression (Fig. 4f), indicative of a NdPr4 fate. Importantly, caudal explants additionally elongated and extended cell protrusions characteristic of ND tip cells[35] (Figs. 2e, 4f NdPr4 panel and Supplementary Fig. 3a).

**Fig. 2 Progenitor lineages in the nephric duct. a** t-distributed stochastic neighbor embedding (tSNE) analysis of nephric duct progenitor (NdPr) cells identified four cell clusters (labeled NdPr1, NdPr2a/b, NdPr3 and NdPr4). **b** tSNE analysis for the expression of nephric duct (*Gata3*) and mesonephric tubules (*Wt1*) lineages markers. **c–e** Main cell clusters markers and validation of spatial and temporal dynamics of the progenitor cell subpopulations in tissue sections. **c** Top 20 signature markers for each cell type shown in (**a**). The negative log10 adjusted *P* values is shown for each marker. Source data are provided as a Source Data file. **d** tSNE analysis for the expression of selected cluster marker. **e** Upper panel: representative picture series-immunofluorescence staining for the selected cluster marker on nephric duct sections at different regions along the anterior-posterior axis of the nephric duct: Rostral, Intermediate (Interm) I, II and Tip. *n* = 5 independent experiments. Lower panel: representative picture series at different developmental stages in the rostral region for NdPr1, NdPr2 and NdPr3 panel, and caudal region for NdPr4. *n* = 5 independent experiments. White and yellow dashed lines denote nephric duct (ND) and mesonephric tubule (MT) cells, respectively. IM: Intermediate mesoderm. Scale bar 25 µm and the same for all the pictures on the same row.

Together, these results suggest a bipotential nature for NdPr2 cells and support a temporal progression of ND progenitor cell types in the urinary tract primordium.

**Gata3 regulates nephric duct progenitor cell progression**. Germline inactivation of *Gata3* leads to defects in ND elongation resulting in the absence of definitive kidneys (renal agenesis)[33,34] (Fig. 5a and b). The single cell RNA-seq experiment from wild type embryos shows that *Gata3* is most highly expressed in the NdPr4 population (Supplementary Fig. 2a and 8c). In light of the progenitor cell populations, we revisited the *Gata3* mutant phenotype to determine whether it was the result of defects in NdPr cell progression. At the morphological level, *Gata3* knockout embryos showed a 40% reduction in elongation at E9.5, in reference to trunk length (Fig. 5c). At E10.5, the *Gata3* mutant NDs were aberrantly shaped, with increased lumen size and number of cells per duct section (Supplementary Fig. 9a, b, c). Interestingly, some ND cells detached from the duct in the mutant (yellow arrow in Supplementary Fig. 9b and d). These cells lost their typical epithelial morphology and showed aberrant expression of the epithelial marker E-cadherin and the apical Par complex protein aPKC (Supplementary Fig. 9d).

We next assessed the cellular signature of *Gata3*-deficient embryos. For this, we performed single cell RNA-seq analysis of *Pax2* expressing cells from control (*Pax2-GFP*) and *Gata3* mutant embryos (*Pax2-GFP; Gata3* KO) at E9.5 (Fig. 5 and Supplementary Fig. 9e). The analysis yielded a total of 2588 cells in control and 3626 cells in *Gata3* mutant embryos. Of these, a total of 367 (control) and 401 (*Gata3* KO) cells had a ND identity. In line with our previous single cell RNA-seq analysis (Fig. 1), *Pax2-GFP* trunk cells clustered into distinct cell populations (Fig. 5d and Supplementary Fig. 9f). tSNE analysis showed a near-perfect overlap among control and *Gata3* mutant in all cell types, except for the ND, where a subpopulation was clearly missing in the *Gata3* mutant sample (Fig. 5e). To explore this intriguing result, we repeated the clustering analysis with only ND cells. As expected, control cells clustered into NdPr1, NdPr2, NdPr3 and NdPr4 (Fig. 5f). In contrast, *Gata3* mutant cells were scarce within the NdPr2 cluster, and completely missing in the NdPr4 cluster (Fig. 5f). Interestingly, a large proportion of mutant cells adopted a hybrid NdPr1/NdPr2 identity (denoted by red line in Fig. 5f). Analysis of the most significantly expressed genes for control and *Gata3* mutant clusters confirmed the lack of NdPr4 gene signature (Fig. 5g, Supplementary Fig. 9g, Supplementary Data 2), which was corroborated by Aldh1a3 staining on tissue sections (Fig. 5h). This analysis also showed that, whereas wild type NdPr1 and NdPr2 clusters are well segregated, most *Gata3*−/− NdPr2 cells still express the main markers of NdPr1 identity, notably *Sfrp2* and *Pcp4* (Fig. 5g). Indeed, the proportion of *Gata3* mutant ND cells expressing NdPr1 markers (*Pcp4*, *Hoxb9* and *Sfrp2*) was increased (Fig. 5h, Supplementary Fig. 9g). The intermediate mesoderm and mesonephric tubule markers *Col1a2* and *Wt1* were not expressed in *Gata3* mutant ND cells,

indicating that mutant cells did not transition to these alternative fates (Fig. 5h, Supplementary Fig. 9g). *Tfap2b* expression in *Gata3* mutant embryos additionally argues against a transcriptional control of *Tfap2b* by Gata3 (Fig. 5g and Supplementary Fig. 9h). Pseudotime trajectory analysis confirmed that *Gata3* mutant cells failed to transition toward a NdPr4 fate (Fig. 5i and Supplementary Fig. 9i). Taken together, these results support a role for Gata3 in the downregulation of the NdPr1 fate in NdPr2 cells and demonstrate a strict requirement for the specification of the NdPr4 fate. The lack of NdPr4 cells is consistent with the absence of UB and metanephric kidneys in *Gata3* mutant embryos.

**Tfap2a/2b in ND morphogenesis and lineage progression**. To better understand the regulatory mechanisms of progenitor cell progression, we focused on NdPr2 cells. As this population is characterized by the expression of *Tfap2b* (Fig. 2c, Supplementary Fig. 2), we hypothesized that Tfap2 family members may act as regulators of this cell type. Single cell RNA-seq data indicated that *Tfap2a* and *Tfap2b* are both expressed in the ND and showed a similar pattern of expression, whereas other Tfap2 members are not significantly expressed (Supplementary Fig. 3a and Supplementary Fig. 10a). This raises the possibility of functional redundancy between both genes.

To study the function of *Tfap2* genes in ND lineage progression, we generated *Tfap2a;Tfap2b* double mutant embryos using a transient zygotic CRISPR/Cas9 inactivation strategy. Zygotic inactivation was achieved by injection of 4 sgRNA per *Tfap2* gene in one cell stage embryos obtained by in vitro fertilization of wild type oocytes with *Pax2-GFP*-derived sperm (Fig. 6a, Supplementary Fig. 10b). Embryos from surrogate mothers were screened for ND phenotypes at E9.5 and E10.5 by wholemount Pax2-GFP fluorescence.

Out of 5 injection rounds, we recovered a total of 36 embryos with inactivating mutations in *Tfap2a* and/or *Tfap2b* genes (9 embryos at E9.5 and 27 at E10.5) (Supplementary Fig. 10c and Supplementary Data 3). *Tfap2a* single or compound *Tfap2a*−/−; *Tfap2b*+/− knockout embryos did not exhibit overt ND phenotypes (Fig. 6b lower panel), but presented open cranial neural fold defects in high frequency (90%) as previously reported[60] (Supplementary Fig. 10d). On the other hand, 85% of compound *Tfap2b* knockout embryos exhibited severe ND integrity defects at E10.5 (Fig. 6b lower panel). Wholemount Pax2-GFP fluorescence revealed severe renal defects in double knockout embryos, with the ND appearing shorter and wider at E9.5 (Fig. 6b upper panel). To quantify the ND phenotype, we compared ND length in control and *Tfap2a/2b* double mutants at E9.5 (Fig. 6c), which revealed a significant 20% reduction in *Tfap2a*−/−;*Tfap2b*−/− ND length. This elongation defect was confirmed in other *Tfap2a/2b* double mutants by immunostaining for the ND marker Gata3 (Fig. 6d). An increase in the number of Gata3+ cells was seen in the rostral region of *Tfap2a/2b* mutant NDs (Fig. 6d and e), which is in line with the increased width of the ND seen by wholemount GFP fluorescence (Fig. 6b upper

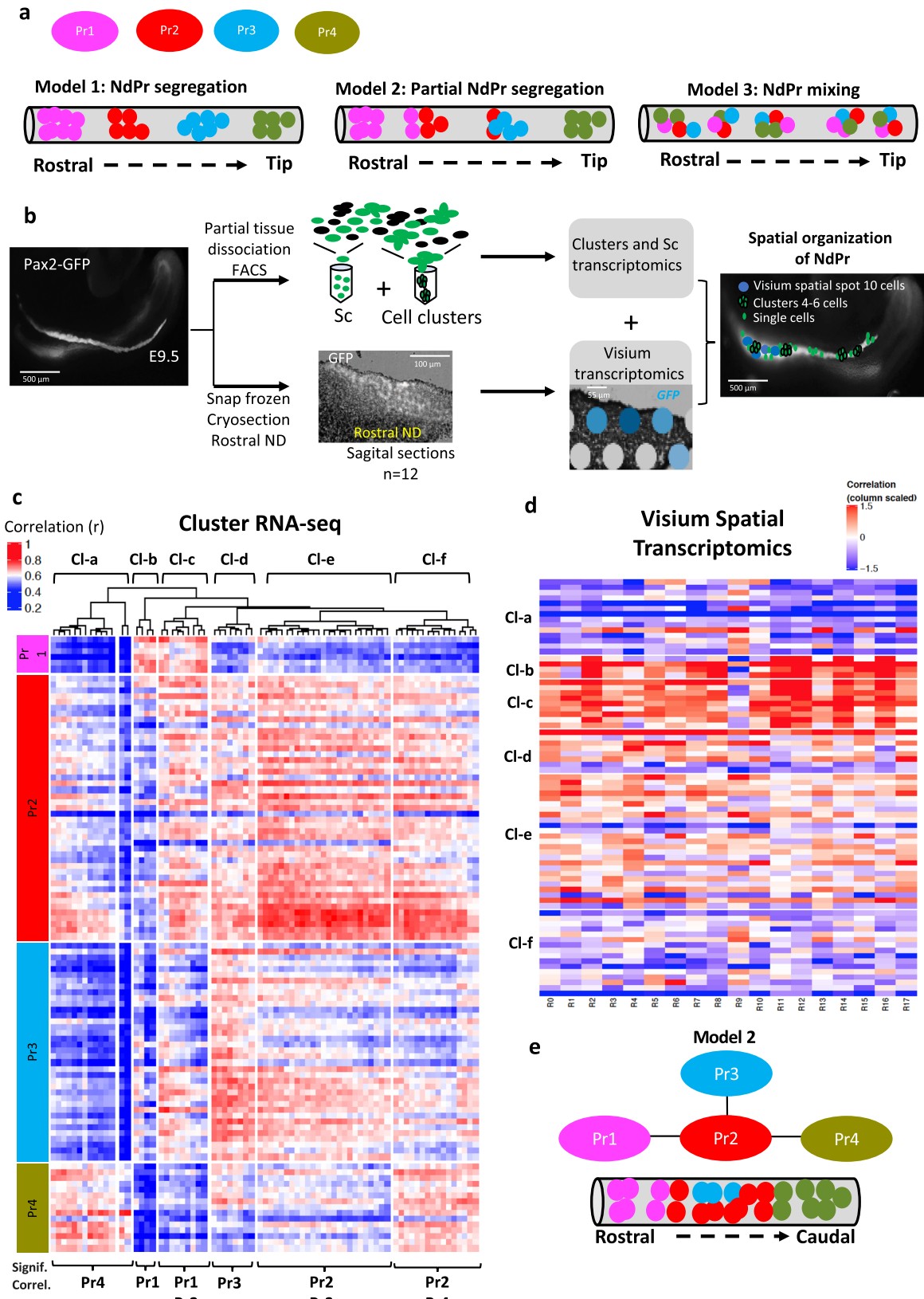

**Fig. 3 Cluster-RNA-seq analysis supports the spatial segregation of nephric duct progenitors. a** Schematic representation of the nephric duct depicting the spatial location of progenitor (Pr) populations according to three different models. Cell identities are color-coded. **b** Schematic representation of Cluster RNA-seq and Visium 10× spatial transcriptomics procedures. **c** Pearson correlation heatmap between Cluster and Single cell libraries, nephric duct lineage only. Single cells are color coded by nephric duct progenitor (NdPr) identity NdPr1, NdPr2, NdPr3 and NdPr4. A summary of the main NdPr identity of Cluster subgroups is shown at the bottom. **d** Pearson correlation heatmap between nephric duct Rostral spatial and Cluster RNA-seq libraries. **e** Proposed model for the spatial organization of NdPr in the nephric duct based on the Cluster RNA-seq analysis. Sc: single cell. Cl: cluster.

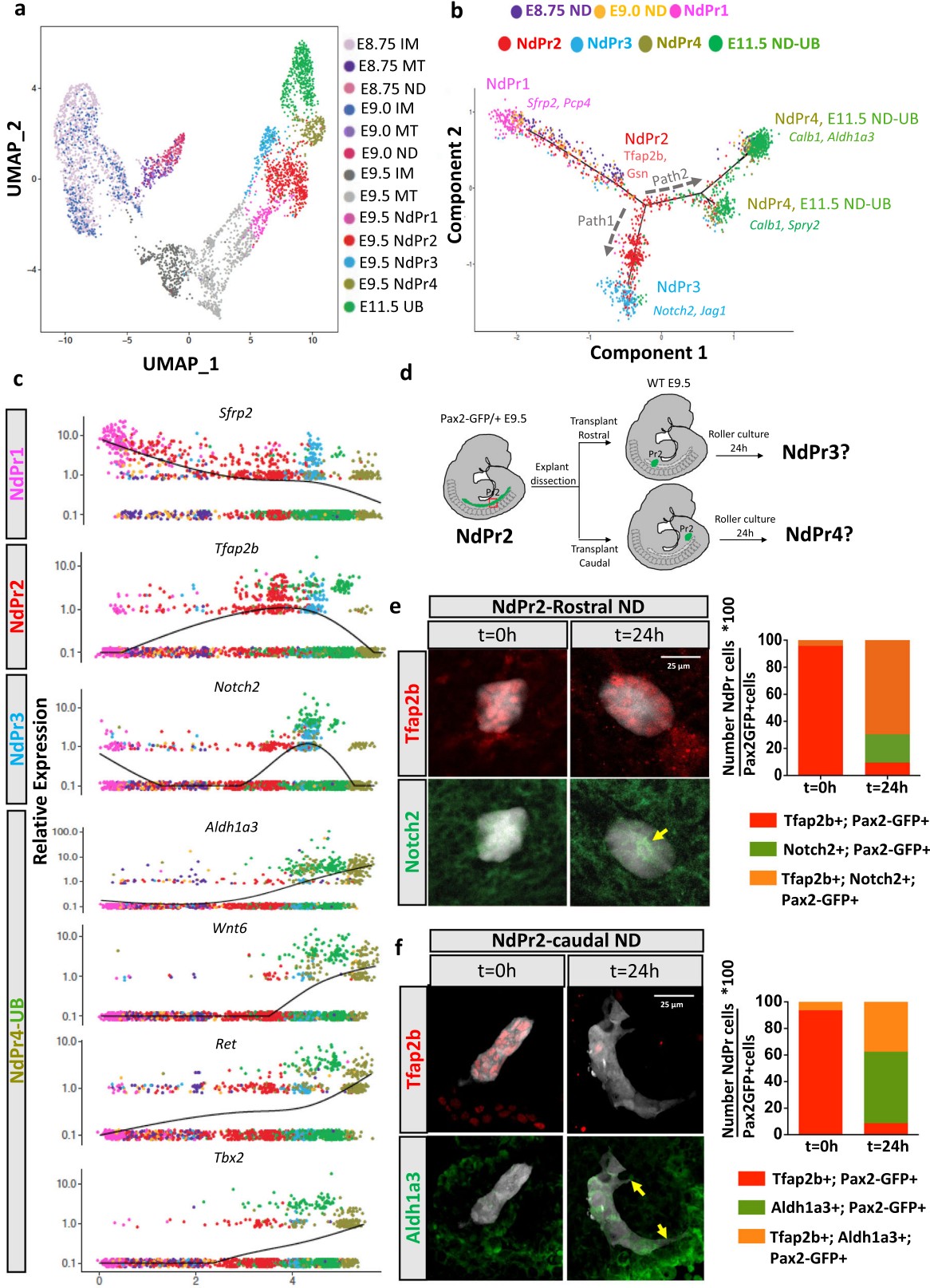

panel). This result further indicates that Gata3 expression does not require *Tfap2a/b* in the ND. Importantly, ectopic Pax2-GFP + or Gata3+ cells were also observed in the adjacent intermediate mesoderm and MT regions (yellow arrow in Fig. 6b and d). To determine whether *Tfap2a/2b* deficient ND cells acquired a different identity, we evaluated the expression of Wt1 and Hoxb9 which mark intermediate mesoderm/MT and NdPr1 cell identities. The number of Gata3+, Hoxb9+ (NdPr1) and Gata3+, Wt1+ double positive cells was significantly increased in the ND of *Tfap2a−/−*;*Tfap2b−/−* embryos (Fig. 6f, g and h), suggesting that *Tfap2a/b*-deficient cells

**Fig. 4 Developmental trajectory of nephric duct progenitors by single cell RNA-seq and transplantation assays. a** Uniform Manifold Approximation and Projection (UMAP) analysis of intermediate mesoderm (IM), mesonephric tubules (MT), nephric duct (ND, NdPr) and ureteric bud (UB) single cell RNA-seq libraries at E8.75, E9.0, E9.5 and E11.5. Cells are color coded by time point and cell type. **b** Principal component analysis plot identifies state transitions between nephric duct progenitor (NdPr) populations at different time point during ND development and the trajectory of cell type differentiation (depicted by black arrows). The trajectory was constructed based on the top 10 NdPr markers. **c** Representative expression profile of the selected NdPr cluster markers in pseudotime. Cells are color coded by NdPr type and time point, as in (**b**). **d** Schematic representation of the tissue transplantation procedure used to assess the developmental potential of NdPr2 cells. **e** Analysis of the developmental potential of NdPr2 rostral grafts. Left panel shows immunofluorescence staining of the NdPr2 marker Tfap2b and the NdPr3 marker Notch2 in rostral grafts (Pax2-GFP positives) at $t = 0$ h and 24 h after transplantation. Right panel shows the quantification of the percentage of NdPr2 (Tfap2b + ), NdPr3 (Notch2 + ) or double positive cells in $n = 3$ ($t = 0$ h) and $n = 3$ ($t = 24$ h) explants. Source data are provided as a Source Data file. **f** Analysis of the developmental potential of NdPr2 caudal grafts. Left panel shows immunofluorescence staining of the NdPr2 marker Tfap2b and the NdPr4 marker Aldh1a3 in caudal grafts (Pax2-GFP positives) at $t = 0$ h and 24 h after transplantation. Right panel shows the quantification of the percentage of NdPr2 (Tfap2b + ), NdPr4 (Aldh1a3 + ) or double positive cells in $n = 5$ ($t = 0$ h) and $n = 4$ ($t = 24$ h) explants. Scale bar 25 µm for all pictures.

---

fail to fully exit the intermediate mesoderm state. Together, these results identify Tfap2a and Tfap2b as regulators of ND morphogenesis and suggest that they act coordinately in lineage progression to promote the transition to NdPr2, by downregulating intermediate mesoderm cell identity.

## Discussion

The kidney initially develops from two major cell populations: the metanephric mesenchyme and the UB lineage. Metanephric mesenchymal cells give rise to nephrons and stromal tissues. This lineage has been widely explored in recent years, notably with the identification of nephron progenitor cells[3–5,7–15,61]. On the other hand, the collecting duct system of the kidney is generated by the UB, which itself is derived from the ND. Hence, the origin of the renal collecting duct system comes down to understanding the progression from ND induction to the UB lineage. However, the ND is still poorly understood at the cellular level. In this study, we have used scRNA-seq and spatial transcriptomics to study the cellular heterogeneity of the ND. We show that the ND is made of distinct cell populations that emerge in a coordinated way to generate the UB precursor cell population (Fig. 7). We demonstrate that Gata3, an established regulator of ND morphogenesis, is a critical regulator of this lineage progression and is strictly necessary for the formation of NdPr4 progenitors (Fig. 7b). Our results additionally identify Tfap2a and Tfap2b as critical players in ND progenitor cell homeostasis through the repression of the intermediate mesoderm fate (Fig. 7b).

The ND develops from anterior intermediate mesoderm precursors that transition to the ND lineage and extends caudally toward the cloaca to form the UB at the level of the hindlimb[4–6]. Our data adds important understanding to these events, both in terms of developmental mechanisms and for guided differentiation of functional kidney tissue. Previous models of ND morphogenesis assumed a structure made of a single cell type showing differences in gene expression levels along the rostro-caudal axis[5,35,62]. Here, we identify the ND as heterogenous at the cellular level, counting four distinct progenitor cell types. Those progenitors segregate in time and space, such that each population is relatively homogenous with regions of mixed identity between subsequent progression stages (Fig. 7a). We show that the NdPr1 state appears early during ND lineage progression. Over time, ND progenitors downregulate the NdPr1 identity to upregulate a NdPr2 fate. At the difference of NdPr1 cells, which retain some intermediate mesoderm identity, the expression of several NdPr2 markers (including Tfap2b, Cited1 and Gsn) is restricted to ND cells.

Our results suggest a developmental pathway whereby NdPr2 develop into two major cell fates: NdPr3 and NdPr4. These include alignment of scRNA-seq data at different developmental stages, Cluster and Visum RNA-seq, marker analysis and tissue

transplantation studies. The NdPr2 to NdPr4 transition is strongly supported by the Gata3 KO scRNA-seq data, where mutant cells accumulate a NdPr1/NdPr2 mixed identity and fail to generate a NdPr4 fate. However, the precise origin of NdPr3 is more difficult to ascertain, as Gata3 KO embryos have both Mt and NdPr3 cells. Our data suggest that NdPr3 cells (Gata3 + ; Tfap2b + ;Wt1–) are mesonephric CnS and can develop from NdPr2. However, in the metanephros (adult kidney), the CnS was shown to derive mostly from nephron progenitor cells expressing Gata3 rather than from collecting duct cells[63]. At this point, it is therefore not possible to exclude that NdPr3 cells derive from the mesonephric tubule lineage (i.e., MtPr3-like cells expressing key ND regulators such as Gata3 and Tfap2b). Alternatively, mesonephric CnS could be of mixed origin, whereby the ND contributes NdPr3 cells and MT contribute MtPr3 cells, as suggested by our results. Formal lineage tracing experiment would be necessary to clarify the lineage hierarchy leading to NdPr3 cells.

Several lines of evidence suggest that NdPr4 is the bona fide UB precursor cell. (1) It is located in the caudal ND at E9.5, but also at E10.5 when and where the UB emerges. (2) The gene signature of NdPr4 is closest to E11.5 UB as shown by trajectory analysis and UMAP. (3) The NdPr4 signature include genes that are strictly required for UB formation, notably Ret[64] and Npnt[51]. Our findings thus reveal intermediate states between anterior intermediate mesoderm progenitors and ND progenitor cells poised for UB induction (Fig. 7b).

Elucidating the lineage relationship between NdPr cells was partly based on a combination of single cell RNA-seq, sequencing of dissociated cell clusters (Cluster-seq) and spatial transcriptomics. The introduced Cluster-seq approach utilizes partially dissociated tissue that retains the spatial relationship between cells but without the need for application of laser capture microdissection as in Smart-3SEQ[53,54], and at higher resolution than RNA-seq from cryosections as used in TOMO-seq[55,56]. Additional spatial transcriptomic experiments using Visium methodology[57,58] validated the Cluster-seq results and confirmed the utility of the method as an affordable approach that can be easily incorporated as part of a single cell capture experiment using the Chromium system. Some caveats nonetheless need to be considered in the analysis of cl-RNAseq data. As the approach is based on the partial dissociation of cells by trypsinization, cells with stronger cell-cell interactions are expected to be overrepresented in cl-RNAseq clusters. In addition, although our results showed clearly defined clusters devoid of cell contaminants, one cannot exclude the possibility of artefactual cell-cell associations from "sticky" cells in the generation of cell clusters. These points should be an integral part of the analysis of cl-RNAseq results.

We previously identified Gata3 as an important regulator of ND morphogenesis[33,35]. In humans, GATA3 is associated with HDR syndrome[65], which is part of the CAKUT spectrum of

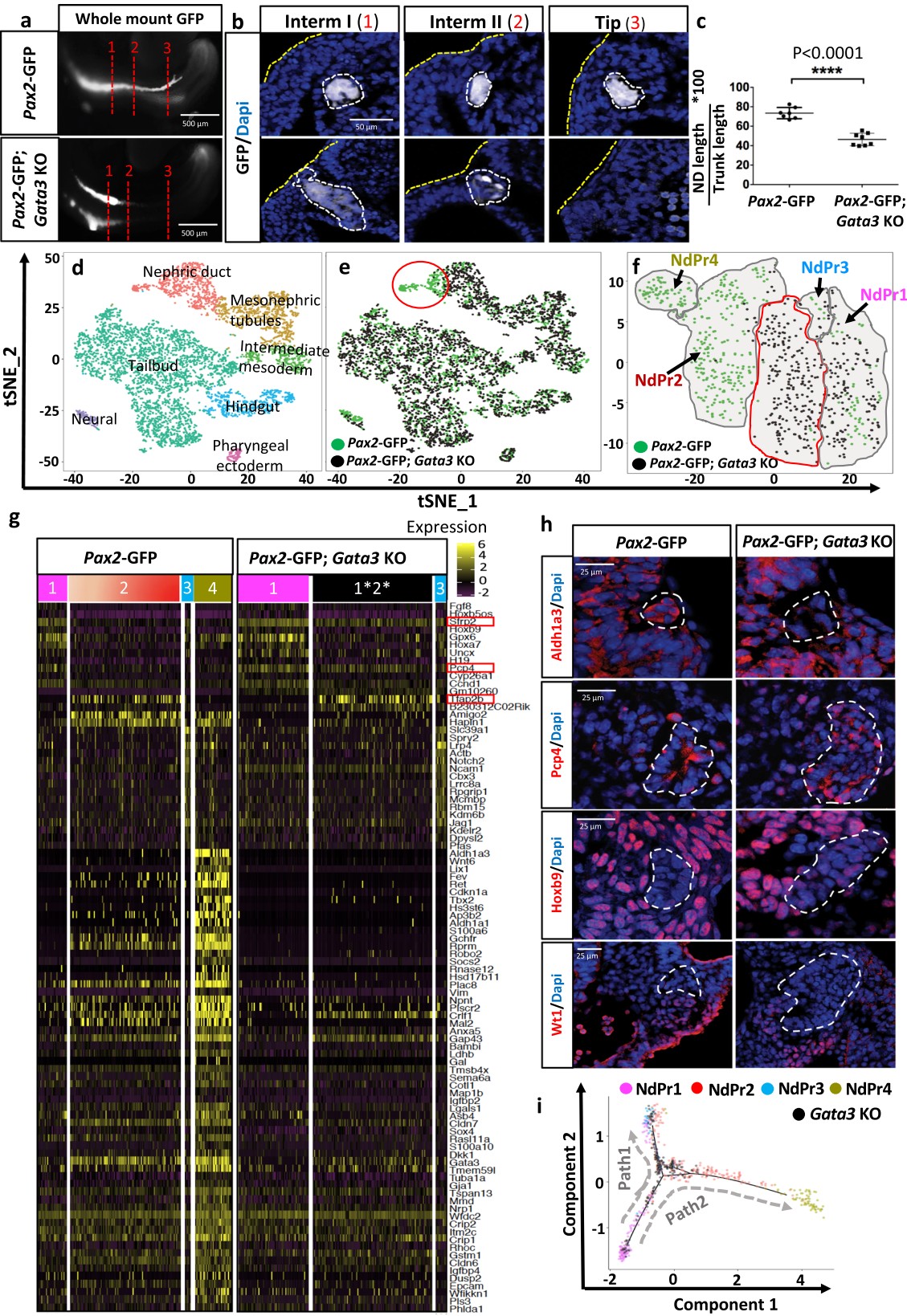

urinary tract diseases[66,67]. *Gata3* deficiency in the mouse results in bilateral renal agenesis accompanied by impaired ND elongation[33]. Conditional inactivation of *Gata3* to bypass these severe defects in early elongation affects the final elongation process and insertion of the ND into the cloaca, resulting in CAKUT-like phenotypes[35]. The mechanisms by which Gata3

plays these important roles are not well understood. This report demonstrates that Gata3 functions in ND development by controlling progenitor cell lineage progression. Most notably, *Gata3* deficient embryos lack the NdPr4 cell fate from which the UB forms. Interestingly, NdPr3 is still specified in the mutant, suggesting that Gata3 may act specifically on the NdPr2-NdPr4

**Fig. 5 Gata3 deficient nephric ducts lack NdPr4 and are arrested at NdPr1-2 stage. a–c** Gata3 loss of function affects nephric duct elongation. **a** Wholemount GFP fluorescence in control (*Pax2-GFP*) and *Gata3* KO embryos (*Pax2-GFP;Gata3* KO) at E9.5. **b** Imaging of nephric duct (*Pax2-GFP* expression) in transversal sections of the embryos shown in (**a**) at different levels: Intermediate (Interm) I, II and Tip, denoted by red dotted lines 1,2,3 in (**a**). Yellow dotted lines denote the ectoderm whereas white dotted lines denote the nephric duct. Scale bar 50 µm for all the pictures. **c** Quantification of nephric duct (ND) length corrected to trunk length in control (*n* = 7) and *Gata3* KO (*n* = 8) embryos at E9.5. The graph represents mean ± SD, compared by two-tailed unpaired *t*-test. Source data are provided as a Source Data file. **d–e** t-distributed stochastic neighbor embedding (tSNE) analysis of single cell-RNA seq libraries from control (*Pax2-GFP*) and *Gata3* KO FACS sorted GFP-positive cells from E9.5 embryos (two libraries superimposed). Libraries in (**d**) are color coded by cell type, whereas in (**e**) they are color-coded by genotype: control (green cells) and *Gata3* KO (black cells). The red circle in (**e**) denotes a cell population lost in the *Gata3* mutant. **f** tSNE analysis of superimposed control (green) and *Gata3* KO (black) nephric duct cells. NdPr clusters are denoted by black arrows. The red line denotes *Gata3* mutant cells with hybrid NdPr1/NdPr2 identity. **g** Heatmap of most relevant cluster markers in control and *Gata3* KO cells, identified using a logFC > 0.25 and an adjusted *P* value < 0.05. Relevant genes are denoted in a red box. **h** Immunostaining analysis in control and *Gata3* KO nephric duct sections at E9.5 for the main markers of population NdPr1, NdPr4 and mesonephric tubule cells. White dotted lines denote the nephric duct. Scale bar = 25 µm for all pictures. **i** Principal component-based trajectory analysis of control and *Gata3* KO cells. The graph shows the trajectory analysis in reference to progenitor subpopulations.

differentiation branch. In addition, *Gata3* mutant NDs successfully generate the NdPr1 identity, but fail to completely adopt a NdPr2 fate, suggesting that Gata3 also acts on the NdPr1 to NdPr2 transition by repressing NdPr1 identity (Fig. 7b).

This work further identifies Tfap2a/2b as regulators of ND morphogenesis. Both TFAP family members are associated with human kidney diseases. *TFAP2B* has been linked to polycystic kidney disease and controls apoptosis of collecting duct and distal tubular epithelial cells in the metanephric kidney[68–70]. *TFAP2A* is differentially expressed in the median/distal tubule compartment of human nephrons[14] while mutations in *TFAP2A* lead to branchio-oculo-facial syndrome and have been associated with multicystic dysplastic kidney[71,72]. *Tfap2a* also play redundant roles with *Tfap2b* during pronephros differentiation in zebrafish[73,74]. Here we show that Tfap2a/2b are required to downregulate the intermediate mesoderm fate in the renal lineage to coordinate ND morphogenesis. It is important to note that *Tfap2a/b* gene inactivation generate one of the most severe renal phenotypes ever described in the mouse[19]. The fact that Tfap2b is a specific marker of the identified NdPr2/3 cell populations testifies of the crucial importance of these cell types for normal kidney morphogenesis.

Although the ND defects of *Tfap2a/2b* and *Gata3* mutant embryos are similar, both transcription factors seem to act through different mechanisms. An upregulation of Wt1 was observed in *Tfap2a/2b*, but not *Gata3* mutant ND cells, indicating that Tfap2a/2b have a more prominent role in repressing the intermediate mesoderm fate in the ND lineage. Conversely, Gata3 seems to have its crucial role later in NdPr2 to NdPr4 cell fate transition while also acting on the complete downregulation of intermediate mesoderm markers in the ND lineage. Tfap2a/2b and Gata3 are also activated independently in the ND, as they are still expressed in each other's mutant background. The molecular interplay between Gata3, Tfap2a/2b and their respective transcriptomes will be important to explore in future studies.

The progressive generation of cell lineages is an important mechanism for tissue specification and patterning seen in many systems. Here we identify cell types and transcriptional regulators of ND morphogenesis. We propose a model of the ND lineage progression that needs to be refined in future studies. Among the remaining questions to be answered are the precise composition and origin of mesonephric CnS in terms of MtPr3 and NdPr3 cell lineages and the gene regulatory network driving NdPr cell state progression. As the lineage progression of the ND is better understood it will inform us on potential new disease genes related to CAKUT and will facilitate the optimization of guided differentiation protocols to generate functional kidney tissue from stem and progenitor cells. The generation of functional renal tissues for organ replacement has become a real possibility with

the impressive progress in the development of kidney organoids from human ES or iPS cells[5,75,76]. However, generating the UB and branching collecting duct component of those renal organoids has been challenging[5,76]. A recent study used the surface markers c-kit and Cxcr4 to isolate and characterize ND progenitor cells for the purpose of improving the generation of UB and branching collecting ducts from human ES cells[5]. Interestingly, *c-kit* is enriched in a large fraction of both NdPr2 and NdPr4 cells (Supplementary Fig. 7a), suggesting that this experiment captured the two key progenitors of ND development. Similarly, UB branching morphogenesis was obtained from human iPS cells induced to become ND (most likely the NdPr4 cell type described here)[76,77]. The identification of a more defined progenitor sequence leading to NdPr4, notably through the directed formation of the NdPr2 intermediate, may facilitate the generation of robust UB and collecting duct tissues for the purpose of renal organ replacement.

## Methods

### Animal studies

*Ethics statement.* All animal experiments performed in this study were conducted in compliance with the Canadian Council of Animal Care ethical guidelines and were approved by the McGill Animal Care Committee. Mice were housed in autoclaved cages with free access to food and water, as well as appropriate and sufficient nesting and bedding material. Mice had a 12 h cycle of light and darkness. Mouse rooms and cages were well ventilated and kept at a temperature range of 20–24 °C, with a relative humidity of 45–65%.

*Experimental mice and embryos.* Pax2-GFP BAC transgenic mice were generated as previously described[44]. *Gata3* KO mice containing a deletion of *Gata3* exon4 and an insertion of an *Ires-GFP* minigene were previously described[33] and maintained in a heterozygous background (*Gata3*+/−). *Pax2-GFP* and *Pax2-GFP;Gata3*−/− embryos (C57Bl6/C3H mixed background) were generated by natural mating. Noon of the day of vaginal plug detection designated E0.5.

*Embryo manipulation and tissue transplantation assays.* Allelic combinations of *Tfap2a/Tfap2b* transient knockout embryos in a *Pax2-GFP* heterozygous genetic background were generated via CRISPR/Cas9 technology as previously described[78] following a multiple sgRNA approach similar to that described in[79]. Briefly, four synthetic crRNA for each gene (Fig. 6a and Supplementary Fig. 10b) spaced 27–105 base pairs were designed following standard design principles to target *Tfap2a* exon 6 and *Tfap2b* exon 4. Same equimolar amount of crRNA and tracrRNA were mixed together and incubated at 95 °C for 5 min to form RNA complexes. RNA complexes were then diluted at 375 ng/ul and mixed individually with Cas9 protein to form RNP complexes. 8 RNP complexes were mixed together so that the final concentration was 50 ng/ul Cas9 protein: 50 ng/ul crRNA:tracrRNA. The RNP cocktail was injected into pronuclear stage zygotes, generated by in vitro fertilization of oocytes from superovulated C57Bl/6 females with sperm from *Pax2-GFP* heterozygous males, to easily visualize ND development by wholemount GFP fluorescence. Transient CRISPR/Cas9 embryos were recovered at E9.5 and E10.5 and analyzed for ND phenotypes.

Tissue transplantation was performed under a SteREO Lumar V12 Carl Zeiss stereomicroscope to visualize, dissect and transplant the donor Pax2-GFP positive ND tissue (Intermediate region) into host wild type E9.5 embryos. Embryos were dissected in DMEM containing 10% fetal bovine serum (FBS) and 1% penicillin/

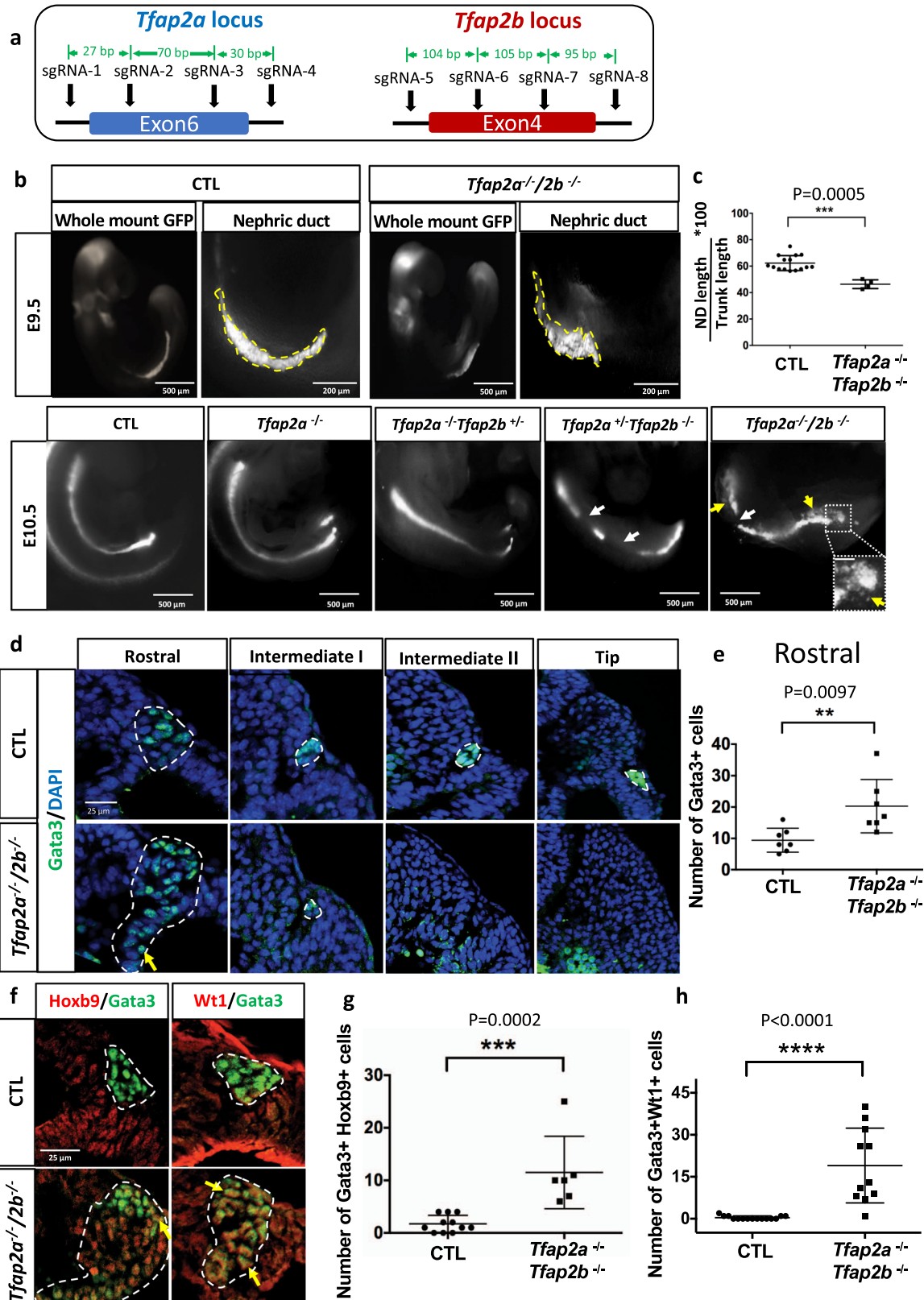

streptomycin. Donor ND tissues were finely dissected with home-made glass capillaries (20 um) to remove surrounding intermediate mesoderm tissue. The dissected tissue was separated in two (for $t = 0$ h and $t = 24$ h conditions) and carefully inserted in the rostral or caudal ND region of the host embryos. The nephrogenic cord and the somites were used as reference points. Rostral explants were inserted above the 10–11 somites, whereas caudal explants were grafted below the tip of the nephrogenic cord. Transplanted embryos were monitored for heart

beating for 30 min and either fixed in PFA 4% (t-0h) or roller cultured for 24 h in DMEM/ 50% rat serum as previously described[46] ($t = 24$ h), and fixed with 4% PFA overnight at 4 °C.

*Mice and embryo genotyping.* All mice were kept in the C57Bl/6 genetic background and genotyped using primers listed in Supplementary Table 1. Genotyping of transient CRISPR/Cas9 embryos was performed by PCR amplification from

**Fig. 6 Inactivation of _Tfap2a_ and _Tfap2b_ by CRISPR/Cas9 leads to defects in nephric duct morphogenesis and progenitor cell identity. a** Schematic representation of _Tfap2a_ and _Tfap2b_ exons targeted by CRISPR/Cas9 technology and the location of the sgRNAs used. **b** Wholemount GFP fluorescence of control (_Pax2-GFP_) and allelic series of _Tfap2a/2b;Pax2-GFP_ mutant embryos at E9.5 and E10.5. White arrows denote nephric duct integrity defects whereas yellow arrow and inset magnification highlight ectopic _Pax2-GFP_ positive cells. Scale bar of the inset = 50 μm. **c** Quantification of nephric duct (ND) elongation in _Tfap2a/2b;Pax2-GFP_ double KO embryos at E9.5. _n_ = 15 (Control) and _n_ = 4 (_Tfap2a/2b_ double KO) biologically independent samples. The graphs represent mean ± SD, assessed by a two-tailed Mann–Whitney test. Source data are provided as a Source Data file. **d** Immunostaining for the nephric duct marker Gata3 in transverse sections of E9.5 _Tfap2a/2b_ double mutant shows an elongation defect and the presence of ectopic Gata3 positive (nephric duct) cells (yellow arrows). Nephric duct cells are denoted by white dotted lines. Scale bar 25 μm for all pictures. **e** Quantification of Gata3+ cells in the rostral region of E9.5 control and _Tfap2a/2b_ double KO embryos (_n_ = 7). The graph represents mean ± SD, determined by a two-tailed unpaired _t_-test. Source data are provided as a Source Data file. **f** Immunostaining for the markers Hoxb9 (NdPr1), and intermediate mesoderm (Wt1) in transversal sections of the nephric duct in E9.5 control and _Tfap2a/2b_ double mutant embryos. Yellow arrows denote Gata3 positive cells expressing Hoxb9 or Wt1 markers. White dotted lines denote the nephric duct. Scale bar 25 μm for all the pictures. **g** Quantification of Hoxb9+,Gata3+ cells number in sections of E9.5 control and _Tfap2a/2b_ double KO embryos. _n_ = 11 (Control) and _n_ = 6 (_Tfap2a/2b_ double KO) independent sections from four different embryos per genotype. **h** Quantification of the number of Wt1+,Gata3+ cells in sections of E9.5 control and _Tfap2a/2b_ double KO embryos. _n_ = 15 (Control) and _n_ = 11 (_Tfap2a/2b_ double KO) independent sections from four different embryos per genotype. The graphs represent mean ± SD, compared by a two-tailed Mann–Whitney test. Source data are provided as a Source Data file.

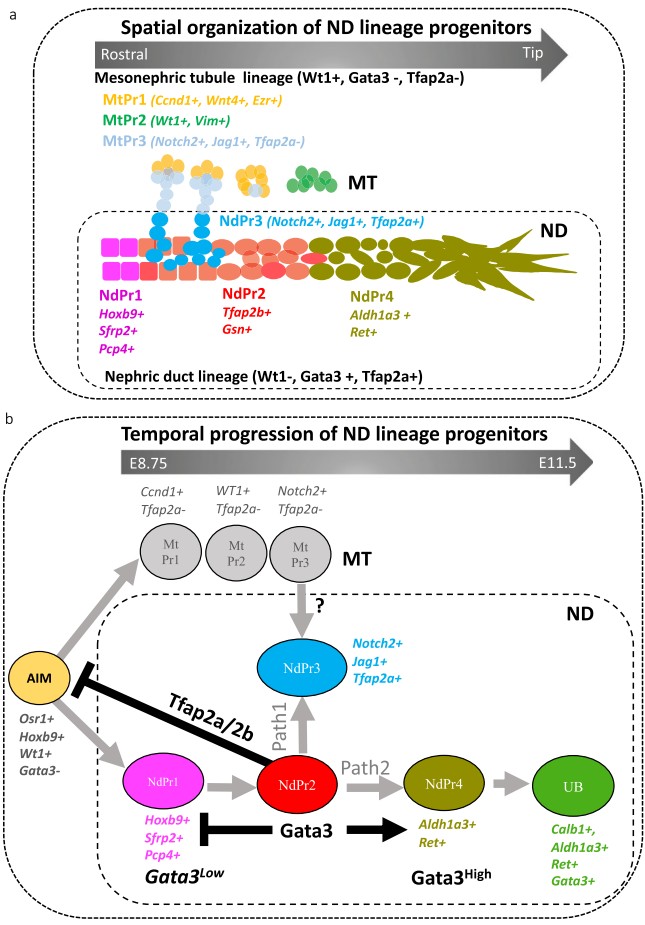

**Fig. 7 Model for lineage progression during nephric duct development.** Schematic representation of the spatial (**a**) and temporal (**b**) hierarchy of nephric duct (ND) progenitor cells. NdPr: nephric duct progenitor, MT: mesonephric tubules, MtPr: mesonephric tubule progenitor, AIM: anterior intermediate mesoderm, UB: ureteric bud.

extracted genomic DNA using primers listed in Supplementary Table 1 followed by Sanger Sequencing and Illumina Miseq PE300 sequencing. Targeting efficiency and analysis of indels was assessed with the web tools Tracking of Indels by Decomposition, Inference of CRISPR Edits (ICE) and Cas-Analyzer and Integrative Genomics Viewer[80–82].

### Single cell and cluster RNA-seq

_Isolation of single cells and cell clusters from the trunk of mouse embryos._ Control (E8.75, E9.0 and E9.5 _Pax2-GFP_; E9.5 _Pax2-GFP;Gata3+/−_; E11.5 C57Bl/6) and

_Gata3_ mutant (E9.5 _Pax2-GFP;Gata3−/−_) embryos were dissected in cold DMEM containing 10% FBS and 1% penicillin/streptomycin. GFP positive single cells were isolated by FACS from dissociated trunk tissue. Material from both control and mutant was collected and processed at the same time. In brief, trunk tissue from stage-matched embryos (3–4 embryos per sample) was dissected and placed in 0.25% trypsin/EDTA at 37 °C for 5 min, with pipette homogenization every 2 min. Fixable Viability dye eFluor 780 staining was added to exclude dead cells. Live intact cells or clusters were FACS sorted using a BD Biosciences FACSAria Fusion machine. Cell clusters were isolated by sorting for higher unit size (FSC-A) (Supplementary Fig. 1a). For isolation of E11.5 caudal urogenital system (UGS), the UGS of 4 wild type stage-matched embryos was dissected out to select the caudal region containing the caudal ND, the UB, as well as the surrounding mesenchyme and the cloaca (Supplementary Fig. 6c). The dissected UGS tissue was digested in 0.25% trypsin/EDTA at 37 °C for 5 min, followed by Collagenase (1 mg/mL) and Dispase (1 in 4 dilution) treatment in DMEM + 10% FBS at 37 °C for 30 min and finally 0.25% trypsin/EDTA at 37 °C for 5 min. Dissociated cells were stained with an anti-Epcam antibody (Biolegend, 118233) and FACS sorted. A mixture of 50% Epcam +ve (epithelial cells including ND, UB and the cloaca epithelium), and 50% Epcam –ve (mesenchymal cells including the metanephric and surrounding caudal mesenchyme) was prepared for library preparation. FACS data was analyzed using Diva (BD Biosciences) version 8.0.2 and FlowJo v10 (BD Biosciences) software.

_Library preparation and sequencing._ Single cells or Clusters were captured using the 10× Genomics Chromium Controller Instrument (10× Genomics, Pleasanton, CA) and ChromiumTM Single Cell 30 Reagent Kits v1 or v2 according to manufacturer's instructions. Briefly, the suspended cells were loaded on a Chromium controller Single-Cell Instrument to generate single-cell Gel Bead-In-Emulsions (GEMs). After breaking the GEMs, the barcoded cDNA was then purified and amplified (14 PCR cycles) on the operative day. Subsequently, the cDNA was fragmented, A-tailed and ligated with adaptors. Finally, ten and 14 cycles of PCR amplification for the single cell and cluster derived libraries was performed to enable sample indexing. The libraries were sequenced on an Illumina HiSeq 4000 or NovaSeq instruments. Information on the number of single cells or Clusters captured, total cDNA yield as well as number of reads per cell or cluster can be found in Supplementary Fig. 1b.

_Single cell RNA-Seq and cluster RNA-seq data analysis._ On average, we achieved >65% of mapping onto annotated genes. Cells/Clusters with low library size, low number of mapped genes or high ratio of reads mapped to mitochondrial DNA and spike-in controls were excluded from the analysis.

Raw sequencing data for each sample was converted to matrices of expression counts using the Cell Ranger software (10× genomics, version 2.0.1). Briefly, raw BCL files from the Illumina HiSeq were demultiplexed into paired-end, gzip-compressed FASTQ files for each channel using Cell Ranger's mkfastq. Using Cell Ranger's count, reads were aligned to the mouse reference transcriptome (mm10), and transcript counts quantified for each annotated gene within every cell.

The resulting UMI count matrix (genes × cells) were then provided as input to Seurat suite version 2.3.4[83]. Cells were first filtered to remove those that contain <200 detected genes and those with >5% of the transcript counts derived from mitochondrial-encoded genes. Sequencing depth of the different libraries was in the range of 2–5 reads per cell per gene (specifically 2.44–5.53), which is very close to the optimal estimate of read per cell per gene to reliably capture markers of cell identity[84]. Data from control and _Gata3_ KO samples were merged into one 10× combined object using canonical correlation analysis (CCA), followed by scaling data (ScaleData function) and finding variable genes (FindVariableGenes function). CCA subspaces were aligned using CCA dimensions 1–20, followed by clustering

(FindClusters function) and integrated t-SNE visualization or Uniform Manifold Approximation and Projection (UMAP) for all cells. Differential expression between control and *Gata3* KO cells, per cluster, was performed using Seurat's FindMarkers.

Single cell and Cluster libraries were merged and analyzed similarly. Cluster libraries were filtered for UMI counts >35,000 and Single cells for UMI counts <9500. This resulted in 777 libraries with clusters and single cells at approximately equal proportions (389 clumps and 388 single cells). Single cell samples were also analyzed separately using the same procedure. For heatmaps, the expression values were normalized per row (z-score).

*Cell cycle and sex genes bias removal.* To mitigate the effects of cell cycling, we calculated cell cycle phase scores based on canonical markers[85], and regressed these out of the data (ScaleData function) (Supplementary Fig.1). Sex-linked genes were removed from the clustering dataset.

### Spatial transcriptomics
*Visium spatial gene expression library preparation and sequencing.* Tissue preparation and slides processing was performed according to the Visium Tissue Preparation Guide (CG000238 Rev A; CG000239 Rev A; CG000240 Rev A, 10× Genomics). Briefly, the trunk of 3 stage-matched E9.5 *Pax2-GFP* mouse embryos was dissected in cold DMEM containing 10% FBS and 1% penicillin/streptomycin, embedded in Optimal Cutting Temperature (OCT) VWR clear frozen section compound and cryosectioned at 10 μm. Some sections were placed in pre-chilled DNA LoBind microcentrifuge tubes 1.5 ml (Eppendorf) for total RNA extraction and assessment of average RNA integrity (RIN) quality score. All samples used had RIN score = 9.5, as defined by 2200 TapeStation (Agilent) with High Sensitivity RNA ScreenTape Assay (Agilent). Tissue sections were permeabilized for 12 min as determined with Visium Tissue optimization procedure. For Tissue Expression, brightfield images were taken on a Nikon Eclipse Ti2 microscope. Raw images were stitched together using NIS-Elements AR 5.11.00 (Nikon) and exported as.tiff files.

Library preparation was performed according to the Visium Spatial Gene Expression User Guide (CG000239 Rev A, 10× Genomics), libraries were loaded and sequenced on a NovaSeq 6000 System (Illumina) as paired-end-dual-indexed with NovaSeq SP PE150 Reagent Kit (Illumina), at a sequencing depth of approximately 25 M read-pairs per capture area. More information on the libraries can be found in Supplementary Fig. 1b. Sequencing was performed using the following run parameters: Read 1, 28 cycles; i7 Index Read, 10 cycles; i5 Index Read, 10 cycles; Read 2, 120 cycles.

*Spatial transcriptomics data analysis.* Raw sequencing data for each sample was converted to matrices of expression counts using the Space Ranger software provided by 10× Genomics (version 1.0). Using Space Ranger's *count*, reads were aligned to the mm10 mouse reference genome, and transcript counts quantified for each annotated gene within every spot. The resulting UMI count matrices (genes × spots) were then provided as input to Seurat suite (version 3.2.0). Spatial datasets were merged, following normalization, variable feature selection, and scaling (SCTransform function). Rostral ND spots were then selected based on the expression of ND markers Pax2 > 0.5 /EGFP > 0.8/Lhx1 > 0.5, positive EGFP signal and right location in the tissue sections.

*Pseudotemporal reconstruction of lineages.* For pseudotime analysis, we applied the R package Monocle 2 version 2.8.0[86]. A single-cell trajectory was constructed by Discriminative Dimensionality Reduction with Trees (DDRTree) algorithm using genes differentially expressed between different clusters. When drawing the heatmap, genes were clustered by their pseudotime expression patterns.

*Pearson correlation analysis for comparison of clusters and single cell signatures.* Correlation heatmap was created by Pearson correlation of a set of differentially expressed genes that marked the different clusters in the relevant single cells, cluster-cells or spatial spots samples. Pearson correlation coefficients were calculated and standardized using the *cor* and *scale* functions, respectively, in R. To determine whether the observed mean correlation is significantly higher in a cluster, we performed random permutation test by shuffling the cluster barcodes 1000 times.

*Gene ontology analysis.* Gene ontology enrichments among differentially expressed genes were obtained using Enrichr web server[87].

### Immunofluorescent staining and image analysis.
For immunofluorescent staining in tissue sections, embryos were fixed in 4% PFA overnight at 4 °C, washed three times in 1× PBS, immersed in 30% sucrose for 24 h, embedded in OCT compound and cryosectioned into 15 μm sections. Slides were washed in 1xPBS, permeabilized in PBS, 0.3% Triton X-100, 0.1% Tween-20 for 7 min at room temperature, blocked with Universal Blocking Reagent solution (BioGenex) for 1 h at room temperature, and incubated with relevant primary antibodies at 4 °C overnight. Primary antibodies and dilutions used are as follows: E-cadherin (1:500, Invitrogen, ECCD-2); Hoxb9 (1:300, Santa Cruz, sc-398500); Tfap2b (1:100, Santa Cruz, sc-390119); Tfap2a (1:100, Santa Cruz, sc-12726); Jag1 (1:100, Santa Cruz, sc-

390177); Notch2 (1:200, Cell Signaling, D76A6), Gata3 (1:100, Invitrogen, 14-9966-82); Spry2 (1:100, LSBio, LS-C499867); Aldh1a3 (1:100, Millipore Sigma, ABN427); WT1 (1:300, Sigma-Aldrich, clone 6F-H2); Ccnd1 (1:100, Abcam, ab134175); Sfrp2 (1:100, Santa Cruz, sc-365524); Pcp4 (1:100, Atlas Antibodies, HPA005792); Kdelr2 (1:100, Santa Cruz, sc-57347), aPKCz/i (1:100, Santa Cruz, SC-216). Sections were counterstained with relevant Alexa Fluor-555 or 647 secondary antibodies (1:500, Invitrogen) or Alexa Fluor-635 conjugated phalloidin (1:40; Invitrogen) and DAPI (50 μg/mL; Invitrogen) for 2 h at room temperature. Wholemount immunofluorescence staining of embryos was performed using the protocol described in[88]. Stained sections or tissues were mounted in ProLong Gold Antifade reagent and imaged on a Zeiss LSM 710 or LSM800 confocal microscope. Images were analyzed with Image J (Fiji) software version 2.0.0-rc-41. Wholemount imaging of mouse embryos was performed on a SteREO Lumar V12 Carl Zeiss stereomicroscope.

### Statistical analyses.
Sample sizes and statistical methods used are described in the figure legends. Statistical analysis was performed by using GraphPad Prism software version 6.0. Difference between means for normally distributed data was tested by an unpaired Student *t* test, whereas for non-normally distributed data was tested by a non-parametric Mann–Whitney test. A *p* value of <0.05 was considered significant.

**Reporting summary.** Further information on research design is available in the Nature Research Reporting Summary linked to this article.

## Data availability
All Figures in the paper (except Fig. 6, Fig. 7, Fig. S3 and Fig. S10) have associated RNA sequencing raw data. All raw RNA sequencing data are publicly available in the Gene Expression Omnibus (GEO) data repository and can be downloaded via the following links: https://www.ncbi.nlm.nih.gov/geo/query/acc.cgi?acc=GSE160136; https://www.ncbi.nlm.nih.gov/geo/query/acc.cgi?acc=GSE160137; https://www.ncbi.nlm.nih.gov/geo/query/acc.cgi?acc=GSE143806; Single cell RNA-seq data from E8.5 embryos was obtained from Marioni lab's website atlas dataset[59] using the link https://marionilab.cruk.cam.ac.uk/MouseGastrulation2018/. Access to other data supporting the findings of this study will be granted upon request to the corresponding author. Source data are provided with this paper.

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

## Acknowledgements

We thank Susanne Kaitna and Matthew Ford for critical reading of the paper and scientific discussions. Special thanks to Yojiro Yamanaka and Rodolphe Soret for experimental help and constructive suggestions throughout the course of this study. We thank members of the Bouchard laboratory for experimental help and critical comments on the paper. Thanks also to the McGill University platforms: Transgenic Core (Mitra Cowan and Nobuko Yamanaka) Advanced BioImaging Facility (ABIF) and Flow Cytometry (Camille Stegen and Julien Leconte), as well as the McGill University and Génome Québec Innovation Centre for their technical support. This work was supported by operating grants from the Kidney Foundation of Canada and Canadian Institutes for Health Research (CIHR PJT-159768). M.B. holds a Senior Research Scholar Award from the Fonds de la Recherche du Québec-Santé (FRQS). O.S.F. was supported by a KRESCENT, MICRTP and Canderel postdoctoral fellowships. Y.Z. was supported by a Faculty of Medicine Internal Studentship.

## Author contributions

O.S.F. and M.B. designed research and wrote the paper. J.R. designed the Cluster RNAseq and spatial transcriptomic analyses as well as supported the single cell RNA-seq experiment. A.P., M.B. and G.B. performed bioinformatic analysis. O.S.F., M.S., Y.C.W. and Y.Z. performed experiments.

## Competing interests

The authors declare no competing interests.
