## [Peer Review File · Nature Communications]

REVIEWERS' COMMENTS

Reviewer #1 (Remarks to the Author):

As a reviewer of this manuscript for a previous Nature stable journal, I have watched the evolution of the story and note the efforts made by the authors to address remaining concerns. The key issues that had to be addressed was the interpretation of the origin of NdPr3, which was equivocal. This has been clearly acknowledged and discussed. The manuscript also now attempts to reconcile their findings with the claims in the field around stem cell-derived UE, for which there is a paucity of transcriptional data. I believe this has resulted in an improved and more balanced manuscript that will be of considerable interest to the field. As such, I believe that this is worthy of publication.

Reviewer #3 (Remarks to the Author):

General:

The authors introduce the claim that this manuscript provides a new way of thinking about the cellular origin of the renal collecting duct system and associated urinary tract developmental diseases. Overall, the manuscript does not provide evidence in support of this claim or the goal outlined in the abstract to facilitate regenerative medicine of the kidney. Additionally, the claim that NdPr4 is the UB precursor cell depends on converging lines of evidence that do not rule out alternative possibilities. The manuscript identifies Tfap2a/2b and Gata3 in coordinating progenitor cell progression, however, the results do not resolve why mutant embryos involving these transcription factors have similar nephric duct defects through different mechanisms. Additional insights into the interactions between Gata3 and Tfap2a/2b would provide additional support for how the progenitors are coordinated during development.

Specific comments:

The authors should discuss the findings from Zeng et al., 2020 describing a 3D branching ureteric bud organoid culture model that was derived from UB progenitors from mouse and human fetal kidneys (<https://www.biorxiv.org/content/10.1101/2020.04.27.049031v1.full>) in the context of their hypothesis for NdPr4.

I suggest that the authors support the extent to which their experimental yield of 3,396 cells in their scRNAseq dataset is sufficient for answering their biological question of interest. The authors should discuss whether their dataset is in line with standards recently discussed within Nature Communications and confirm that a sufficient number of cells was profiled to accurately estimate gene expression. <https://www.nature.com/articles/s41467-020-14482-y> Essentially, the authors should comment on their experimental design by explaining number of cells vs. sequencing depth. Is the cell number sufficient to capture the cell types of interest? Is the sequencing depth appropriate for the biological conclusions being made? These questions should be applied to Figure 5 when assessing the cellular signatures of Gata3-deficient embryos.

Given the author's interest in developing a new strategy to assess the spatial heterogeneity of the different cell populations, Cluster RNA sequencing (cl-RNAseq), I would expect additional supporting

information as to why cell clusters were used as an input instead of single cells for the RNAseq library. How does cl-RNAseq improved upon protocols used to capture cells that are physically connected in other spatial transcriptomics approaches? The authors provide validation of the Cluster-seq results (Figure 3D); however, the advantages of Cluster-RNA-seq over other approaches should be made clear. Theoretical advantages are discussed in the discussion section, but additional justification for introducing a new approach should be made clear in Figure 3.

A major claim of the manuscript is that Tfp2a and Tfp2b act as novel regulators of ND morphogenesis and that Tfp2a/b-deficient cells return to a more primitive, "intermediate mesoderm-like state." Can the authors more robustly confirm the phenotype described in Figure 6E.

Please find below the detailed response to the remaining reviewer's comments.

Reviewer #1 (Remarks to the Author):

As a reviewer of this manuscript for a previous Nature stable journal, I have watched the evolution of the story and note the efforts made by the authors to address remaining concerns. The key issues that had to be addressed was the interpretation of the origin of NdPr3, which was equivocal. This has been clearly acknowledged and discussed. The manuscript also now attempts to reconcile their findings with the claims in the field around stem cell-derived UE, for which there is a paucity of transcriptional data. I believe this has resulted in an improved and more balanced manuscript that will be of considerable interest to the field. As such, I believe that this is worthy of publication.

Reviewer #3 (Remarks to the Author):

General:

The authors introduce the claim that this manuscript provides a new way of thinking about the cellular origin of the renal collecting duct system and associated urinary tract developmental diseases. Overall, the manuscript does not provide evidence in support of this claim or the goal outlined in the abstract to facilitate regenerative medicine of the kidney.

Recapitulating normal development of the UB lineage from stem cells has been a challenge in the field of regenerative medicine of the kidney, for which, as reviewer 1 pointed out, there is still a scarcity of transcriptional data. Our manuscript identifies new cell states in the progression between renal lineage specification from the intermediate mesoderm and the formation of the ureteric bud. Importantly, it provides extensive transcriptional data at different time points during this progression, which is expected to inform, guide and facilitate regenerative medicine efforts.

Additionally, the claim that NdPr4 is the UB precursor cell depends on converging lines of evidence that do not rule out alternative possibilities.

NdPr4 are the result of ND lineage progression and is the first cell type of this progression to share an extensive molecular signature with UB cell (marker analysis, scRNAseq). As UB cells are derived from the ND and NdPr4 ND cells represent the dominant cell type when and where the UB forms, we believe the "converging lines of evidence" are fairly strong in support of our claim.

The manuscript identifies Tfap2a/2b and Gata3 in coordinating progenitor cell progression, however, the results do not resolve why mutant embryos involving these transcription factors have similar nephric duct defects through different mechanisms. Additional insights into the interactions between Gata3 and Tfap2a/2b would provide additional support for how the progenitors are coordinated during development.

We agree that the detailed interplay between Gata3 and Tfap2a/b will be very interesting to understand in more details (as already pointed out in the discussion). However, this manuscript is focused on the identification of the lineage progression leading to UB formation. To demonstrate this progression, we revisited the phenotype of Gata3 (a known regulator of ND morphogenesis) and identified Tfap2a/b as new regulators of ND development. We show that these transcription factors are regulated independently of each other, but the detailed molecular mechanisms linking the two is well beyond the scope of this study. The fact that they have a similar phenotype (defect in ND elongation) but harbour different molecular phenotypes (upregulation or not of IM markers) is an interesting observation but it merely shows that the disruption of early lineage progression has a profound effect on ND morphogenesis.

Specific comments:

The authors should discuss the findings from Zeng et al., 2020 describing a 3D branching ureteric bud organoid culture model that was derived from UB progenitors from mouse and human fetal kidneys (<https://www.biorxiv.org/content/10.1101/2020.04.27.049031v1.full>) in the context of their hypothesis for NdPr4. *The manuscript by Zeng et al nicely identified the culture conditions to maintain UB progenitor cells (UPCs) and to allow their differentiation into CD. They could also achieve the generation of UPCs from hESCs and hiPSCs. However, the generation of UPCs was done with the help of fluorescent markers to purify the cells of interest and the*

transition from Pax2+ progenitor cells to UPCs took about 3 weeks, whereas this transition occurs within hours to days in the mouse and human embryos. As impressive as the work of Zeng et al is, it actually supports our claim that a better understanding of the IM to UB lineage progression is necessary for the more efficient and accurate generation of UPC cells from pluripotent stem cells for the purpose of regenerative medicine. We have added a reference to Zeng et al to the discussion.

I suggest that the authors support the extent to which their experimental yield of 3,396 cells in their scRNAseq dataset is sufficient for answering their biological question of interest. The authors should discuss whether their dataset is in line with standards recently discussed within Nature Communications and confirm that a sufficient number of cells was profiled to accurately estimate gene expression. <https://www.nature.com/articles/s41467-020-14482-y> Essentially, the authors should comment on their experimental design by explaining number of cells vs. sequencing depth. Is the cell number sufficient to capture the cell types of interest? Is the sequencing depth appropriate for the biological conclusions being made? These questions should be applied to Figure 5 when assessing the cellular signatures of Gata3-deficient embryos.

We agree that the question of cell number vs. sequencing depth is very important in the design of scRNAseq experiments. It is also true that this trade-off has to be adapted to the biological question. As argued previously, in our experiments, we sequenced about 2/3 of the TOTAL NUMBER OF CELLS present in the organ. Very few scRNAseq experiments capture an organ to such an extent. It is important to remember that this is an organ anlage containing a few thousand cells and that the renal lineage was not present in the embryo 12-24 hours before our experimental time points. By capturing 2/3 of the whole structure, we identified a completely new cell type (NdPr2) but also intermediates stages between cell types (e.g. NdPr1).

More specifically on the question of cell number vs depth, the Zhang paper mentioned by this reviewer estimates that 1 read per cell per gene is generally an optimal coverage to properly capture cellular heterogeneity. In our experimental samples, we obtained 2-5 reads/cell/gene for our RNAseq libraries. This is slightly above but very close to the optimal sequencing depth identified by Zhang et al to minimize the error and obtain the appropriate gene expression distribution necessary to identify cell populations of interest. We have added this information in the Single cell and Cluster RNA-seq Analysis section of Methods and referenced the Zhang paper.

Given the author's interest in developing a new strategy to assess the spatial heterogeneity of the different cell populations, Cluster RNA sequencing (cl-RNAseq), I would expect additional supporting information as to why cell clusters were used as an input instead of single cells for the RNAseq library.

The introduction of clusters was specifically designed to use the basic 10X scRNAseq approach to obtain information about the heterogeneity and 3D organization of the cell populations present in the sample. We essentially reasoned that the superimposition of single cell and cluster transcriptomic profiling derived from the same experimental procedure (10X scRNAseq) would provide proximity data and therefore 3D and heterogeneity information.

How does cl-RNAseq improved upon protocols used to capture cells that are physically connected in other spatial transcriptomics approaches? The authors provide validation of the Cluster-seq results (Figure 3D); however, the advantages of Cluster-RNA-seq over other approaches should be made clear. Theoretical advantages are discussed in the discussion section, but additional justification for introducing a new approach should be made clear in Figure 3.

Cl-RNAseq is an alternative approach to the spatial transcriptomic approaches developed in recent years. One clear advantage is that it uses the exact same setup used for single cell RNAseq (e.g. 10X Genomics and FACS). It is meant as an extension and complementation to generating scRNA-seq data using the same reagents and allowing a capture in the same Chromium chip along the single cell one(s) from the same sample, thus assuring comparable technical performance. In addition, not all research facilities can afford sophisticated spatial transcriptomics equipment and not all scientists can afford the high cost of some of those approaches (e.g. Visium).

Since the ND is a small and thin tubular structure along the trunk of the embryo, it is challenging to capture all regions of interest. Several (expensive) Visium expression slides are needed to get a representative number of spots targeting all the ND regions. In contrast, cell clusters representative of all regions can be easily obtained by FACS for cl-RNAseq. Moreover, we show that for very small structures, cl-RNAseq is in fact superior to the Visium approach as it prevents contamination from surrounding cells when the cells of interest represent only a subset of the Visium "spot" diameter. The full potential of Cl-RNAseq will likely emerge as more and more people adopt this approach as a simple and affordable mean to obtain 3D/heterogeneity information about their samples. We have added a more explicit justification for the use of cl-RNAseq in reference to Fig. 3.

A major claim of the manuscript is that Tfp2a and Tfp2b act as novel regulators of ND morphogenesis and that Tfp2a/b-deficient cells return to a more primitive, “intermediate mesoderm-like state.” Can the authors more robustly confirm the phenotype described in Figure 6E.

In figure 6, we show that Tfp2a/b double mutant embryos show severe elongation and morphological defects, including presence of ectopic Pax2+Gata3+ cells in the adjacent IM region, and the aberrant expression of IM markers. It is important to note that expression of the intermediate mesoderm marker Wt1 is usually quickly downregulated upon ND formation. The significant number of Gata3+Wt1+ cells in the Tfp2a/2b mutant ND indicates that these mutant cells fail to fully exit the IM stage, which we also corroborated by staining with another IM marker, Hoxb9. For clarity, we have modified the statement “Tfp2a/b-deficient cells return to a more primitive intermediate mesoderm-like state” to “Tfp2a/b-deficient cells fail to fully exit the intermediate mesoderm state” as there is no evidence that the mutant cells ever reached a fully committed NdPr2 state.

This is a first description of this mutant phenotype primarily aimed at demonstrating the importance of the new NdPr2 cell type (for which Tfp2a/b are specific markers). In this context, we believe the current characterization properly achieve the goals of 1- Identifying new regulators of ND development, 2- Demonstrating the importance of NdPr2 cells in ND morphogenesis and lineage progression and 3- Showing that Tfp2a/b are necessary to fully commit to the NdPr2 stage (failed lineage specification).